# Identification and Characterization of Short-Term Motor Patterns in Rest Tremor of Individuals with Parkinson’s Disease

**DOI:** 10.3390/healthcare10122536

**Published:** 2022-12-14

**Authors:** Amanda Rabelo, João Paulo Folador, Ariana Moura Cabral, Viviane Lima, Ana Paula Arantes, Luciane Sande, Marcus Fraga Vieira, Rodrigo Maximiano Antunes de Almeida, Adriano de Oliveira Andrade

**Affiliations:** 1Centre for Innovation and Technology Assessment in Health (NIATS), Faculty of Electrical Engineering, Federal University of Uberlândia, Uberlândia 38400-902, Brazil; 2Neuroscience Department, Hotchkiss Brain Institute, University of Calgary, Calgary, AB T2N 1N4, Canada; 3Neuroscience and Motor Control Labotaroty (Neurocom), Federal University of Triagulo Mineiro (UFTM), Uberaba 38025-350, Brazil; 4Bioengineering and Biomechanics Laboratory, Federal University of Goiás, Goiânia 74690-900, Brazil; 5Systems Engineering and Information Technology Institute, Federal University of Itajubá, Itajubá 37500-903, Brazil

**Keywords:** rest tremor, short-term motor patterns (STMPs), gyroscope, Parkinson’s disease

## Abstract

(1) Background: The dynamics of hand tremors involve nonrandom and short-term motor patterns (STMPs). This study aimed to (i) identify STMPs in Parkinson’s disease (PD) and physiological resting tremor and (ii) characterize STMPs by amplitude, persistence, and regularity. (2) Methods: This study included healthy (N = 12, 60.1 ± 5.9 years old) and PD (N = 14, 65 ± 11.54 years old) participants. The signals were collected using a triaxial gyroscope on the dorsal side of the hand during a resting condition. Data were preprocessed and seven features were extracted from each 1 s window with 50% overlap. The STMPs were identified using the clustering technique k-means applied to the data in the two-dimensional space given by t-Distributed Stochastic Neighbor Embedding (t-SNE). The frequency, transition probability, and duration of the STMPs for each group were assessed. All STMP features were averaged across groups. (3) Results: Three STMPs were identified in tremor signals (*p* < 0.05). STMP 1 was prevalent in the healthy control (HC) subjects, STMP 2 in both groups, and STMP3 in PD. Only the coefficient of variation and complexity differed significantly between groups. (4) Conclusion: These results can help professionals characterize and evaluate tremor severity and treatment efficacy.

## 1. Introduction

Tremor is a rhythmical and involuntary oscillatory movement of a body part that can be physiological, as seen in healthy people, or pathological, as observed in several motor disorders [1]. The physiological tremor has a smaller amplitude and can be present in all subjects. In contrast, a pathological tremor presents a higher amplitude and is associated with disorders such as Parkinson’s disease (PD) [2]. The most prevalent tremor in PD is the resting tremor [3]. This tremor manifests when a voluntary muscle contraction is absent in a specific body segment [3,4]. The duration, frequency, and amplitude of this motor sign can be used to characterize the complexity and severity of the disease.

According to the literature, a physiological tremor has a frequency range of 8 to 10 Hz, whereas the rest tremor in PD occurs in a 3 to 6 Hz range [2]. However, higher tremor frequencies up to 6 Hz can be found in PD patients at the early stage [5], and older people tend to have lower tremor frequencies [6]. Due to these frequency overlaps, a frequency analysis is not enough to discriminate tremors [7].

Understanding the origins of tremors is critical for evaluating and improving therapies for various pathological tremors. The cause of tremor is unknown and has been studied extensively [2,3]. Some authors attribute the tremor to motor unit firing in the peripheral neuromuscular system, central oscillations, or cardio ballistic effects [2]. For instance, the physiological tremor origin is unlikely to be caused by neurogenic distortion. In PD, however, there is no agreement on the cause of tremor [8]. Some studies have linked tremor to oscillating thalamic activity, while others have linked it to an action of the basal ganglia [3].

Tremors can also be characterized based on the affected body region, such as the upper or lower limb [9]. Many studies have focused on assessing hand tremors [10,11,12,13,14], which is the topic of this paper, regardless of the type of tremor, i.e., physiological or pathological. Hand tremors are caused by a number of physiological components, including the activity of the basal ganglia, cerebellar circuits, and peripheral nerves, all of which interact on different spatiotemporal scales. Hand tremor dynamics are complicated because they may involve the existence of nonrandom, short-term motor patterns (STMPs).

Using electromyography, some studies [15,16,17] suggested the presence of STMPs as a result of tremors. For instance, Dietz et al. [15] investigated the characteristics of motor unit activity in individuals with PD. They identified three relevant patterns: rhythmic spontaneous resting discharge, abnormally low firing rates during voluntary contraction, and consistent firing patterns. Similarly, Agapaki et al. [16] focused on detecting and exploring the characteristics of motor unity (MU) synchronization and discharge patterns during rest and postural tremor. Rissanen et al. [17] also analyzed differences in electromyographic patterns between individuals with PD and healthy subjects.

According to these authors [15,16,17], STMPs may be linked to the underlying mechanisms that cause tremors, and STMPs may be present in both healthy people and people with PD. STMPs exhibit selfsimilar structures across multiple scales of time and have a hidden dynamic with underlying structures responsible for the abnormal tremor. The characterization of STMPs differs according to the type of tremor, e.g., physiological or pathological. Once a better knowledge of STMPs is achieved, such information may be used to develop better therapies and follow up on different types of pathological tremor.

To identify and quantify tremors, a variety of instruments and sensors are used. These include electroencephalograms (EEG), magnetoencephalograms (MEG), electromyograms (EMG), inertial sensors, and microelectrode recordings (MER) [18]. Each approach enables tremor investigation from a unique perspective. One way to evaluate the force and muscle contraction associated with tremor is to use muscle activity. Kinetic parameters associated with involuntary movement, on the other hand, are best represented and inferred by angular velocity, linear acceleration, and magnetic field change, which are physical variables acquired by gyroscopes, accelerometers, and magnetometers, respectively. 

In particular, several studies use inertial sensors to measure tremor intensity and the frequency of these oscillations [19,20,21]. To the best of our knowledge, there is a lack of studies attempting to identify and characterize STMPs in pathological tremor in people with PD, specifically using inertial sensors. According to the literature, gyroscopes and accelerometers enable the objective and accurate quantification of this signal [22].

As a result, this study hypothesizes that tremor signals detected by inertial sensors can be used to identify STMPs. Furthermore, the appearance of STMPs is controlled by a hidden dynamic that describes their instantaneous appearance. The following questions directed the execution of this research: (i) Is it possible to identify STMPs in rest tremor? (ii) Are there differences between the STMPs of the healthy control (HC) and PD groups? Is it possible to cluster STMPs and characterize them in terms of the extracted features? 

To support these hypotheses, this study aimed (i) to identify the STMPs present in resting tremor in individuals with PD and physiological tremor and (ii) to characterize STMPs in terms of amplitude, persistence, and regularity.

## 2. Materials and Methods

Figure 1 depicts the main steps employed for the identification and characterization of STMPs in PD.

The subjects were recruited and the data were collected in step 1 (Figure 1) of the experimental protocol. After that, the data set was preprocessed. The signals were then split into windows for feature extraction. The estimated features were standardized before being mapped onto a lower-dimensional space using t-SNE. STMP identification was accomplished through clustering with k means. Once the STMP groups were identified, they were evaluated and classified based on their amplitude, persistence, and regularity.

The Human Research Ethics Committee approved this study. Inertial signals were recorded from 26 subjects. The subjects were divided into the following groups: neurologically healthy control individuals (HC = 12, aged 60.1 ± 5.9 years) and individuals with PD (PD = 14, aged 65 ± 11.98 years) clinically diagnosed by a neurologist. Individuals with PD had no evidence of dementia or musculoskeletal changes in the upper limb that were not related to PD, and they were in the ON state of medication. 

The subjects with PD were recruited at the Parkinson Association of Triângulo (Associação Parkinson do Triângulo, Uberlândia, Brazil), an association for the evaluation and treatment of individuals with PD. Before participation in the experimental protocol, a detailed explanation was given to the participants who voluntarily signed a consent form. 

Item 20 of the Unified Parkinson’s Disease Rating Scale part III (MDS—UPDRs III) was used to assess the resting tremor. The score for item 20 is based on the hand tremor at rest and ranges from 0 to 4, with 0 representing a nonvisible tremor and 4 representing the most severe tremor. A specialist not involved in the data analysis examined the PD subjects clinically. The medication was administered to the subjects between 60 min before the experimental protocol, so the scores were assigned during the ON period of the medication. The UDPRS scores for all PD subjects are shown in Table 1. 

### 2.1. Experimental Setup

The TREMSEN device (Precise Tremor Sensing Technology, National Institute of Intellectual Property—Brazil—BR 10 2014 023282 6) [20] containing one triaxial inertial measurement unit (IMU) was employed for data acquisition and real-time visualization. Each IMU consists of a gyroscope unit (L3G4200D, STMicroelectronics, Geneva, Switzerland) and an accelerometer–magnetometer combined unit (LSM303DLM, STMicroelectronics, Switzerland). The sensitivity settings of the IMU can be changed individually via I^2^C communication. 

The sensitivities of the gyroscope, accelerometer, and magnetometer were set to ±500 dps, ±4 g, and ±2 Gauss, respectively, for all subjects, except subject 14, who had higher UPDRS scores (Table 1). Due to the high severity of the tremor of this individual, the sensitivities were set to ±2000 dps, ±16 g, and ±12 Gauss. All the employed settings were carried out in previous studies [23,24] and were chosen to avoid the saturation of the signal conditioner. Signals were sampled at 50 Hz and digitized by a 12-bit analog-to-digital converter.

The experimental protocol used one IMU placed on the hand of the subjects (Figure 2). Data were collected from the dominant hand of healthy subjects and the most affected side of PD subjects. 

Tremulous activity can occur in all directions, depending on limb placement and movement. Figure 2 presents the axes of orientation, in which X, Y, and Z are the proximal–distal, medial–lateral, and dorsal–palmar axes, respectively. For the hand rest tremor assessment, we only evaluated data from the X-axis of the gyroscope (IMU-1) (Figure 2).

IMU1 was placed on the dorsal side of the hand, between the second and fourth metacarpophalangeal joints. 

The experiments were carried out in a room where the subject sat in a comfortable chair with the forearm fixed on an adjustable support surface, the upper arm positioned alongside the trunk, and the elbow flexed at 90°. This arrangement allows movements only in the wrist joints, with the hand in a relaxed position, as shown in Figure 2. All subjects performed three trials lasting approximately 15 s, with 60 s of resting between trials.

### 2.2. Signal Preprocessing

The signal preprocessing is fully explained in [20]. Briefly,

For a better visualization of the STMP in the tremor signal, the collected data were resampled at a sampling frequency of 300 Hz using splines.The signals were smoothed by applying Tukey’s Running Median Smoothing technique, followed by the removal of low-frequency components related to involuntary movements unrelated to the tremor and linear trends from the signals.Outliers were detected by a visual inspection of the box plots and then removed.All signal preprocessing and statistical analyses were performed using R.

### 2.3. Feature Extraction

Prior to feature extraction, the signals were divided into 1 s windows with 50% overlap (Figure 3). To identify the STMPs in the signal, the following linear and nonlinear features were extracted from each window: mean absolute value (MAV), coefficient of variation (CV), zero crossing rate (ZCR), sample entropy (SampEn), and Hjorth parameters (activity-ACT, mobility-MOB, complexity-COMP), as defined in Table 2. According to [25,26,27], evaluating linear and nonlinear features is desirable to obtain complementary information from hand tremors. As a result of the feature estimation, the data in each signal window were represented by a feature vector of dimension seven.

The primary goal of this study was to detect STMPs and characterize them using the studied time domain features. Although these features are estimated in time, some of them (ZC and Mobility) capture changes in signal frequency. In this sense, the set of features chosen combines the features capable of capturing information about changes in the frequency, amplitude, and predictability of the signals.

### 2.4. Dimensionality Reduction

The dimensionality of the data, i.e., the feature vector, was reduced through t-Distributed Stochastic Neighbor Embedding (t-SNE) [31], which converts data from a high-dimensional space into a low-dimensional space while preserving the stochastic distribution of the data points. The data were standardized before the dimensionality reduction.

t-SNE is a nonlinear method that converts the high-dimensional data set $X=\left\{x_{1},\;x_{2},\;x_{3},\ldots ,\;x_{n}\right\}$ into two- or three-dimensional data $Y=\left\{y_{1},\;y_{2},\;y_{3},\ldots ,\;y_{n}\right\}$. The low-dimensional data Y is represented as a map, while the low-dimensional representations $y_{i}$ of the individual data points are represented as map points. The SNE algorithm converts the Euclidean distances between the high-dimensional data points into conditional probabilities ($p_{ij}$ and $q_{ij}$). The pairwise similarities between two data points, $x_{i}$ e $x_{j}$, are represented by the conditional probability $p_{ij}$ in the original high-dimensional space, and the low-dimensional conditional probability is represented by $q_{ij}$ of points $y_{i}$ and $y_{j}$. For nearby data points, $p_{ij}$ is relatively high, whereas for widely separated data points, $p_{ij}$ will be almost infinitesimal.

In order to determine how much the low-dimensional model well represents the high-dimensional model, the t-SNE minimizes the Kullback–Leibler (KL) [31] cost function using a gradient descent method (Equation (1)).
(1)$$C=\underset{i}{\sum }KL(P_{i}∥Qi)=\underset{i}{\sum }\underset{j}{\sum }p_{ij}log\frac{p_{ij}}{q_{ij}}$$
where $P_{i}$ is the conditional probability distribution over all data points given a data point $x_{i}$, and Qi represents the conditional probability distribution over all other map points given a map point $y_{i}$. The location of the points $y_{i}$ in the map is defined by minimizing the KL divergence of the distribution *P* from the distribution Q. The goal is to optimize the embedding such that $p_{ij}$ and $q_{ij}$ are as similar as possible. t-SNE improves SNE by using the Student’s t-distribution rather than a Gaussian to compute the similarity between two points in the low-dimensional space [31]. 

The perplexity parameter in t-SNE determines each point’s optimal number of close neighbors. Van der Maaten and Hinton [31] suggest perplexity values between 5 to 50. However, due to the number of points from our sample, the algorithm did not properly work with perplexity values higher than 15. Then, we ran some experiments with perplexity values in the range between 5 and 15 to verify how this parameter affected the quality of the generated maps. With the map generated by t-SNE, we observed that the perplexity values smaller than 10 generated a large number of clusters dominated by local variations. In contrast, with perplexity values higher than 10, the number of clusters was reduced. Based on this, the perplexity value was set to 10.

### 2.5. Identification of STMPs

STMPs were identified for the HC and PD groups using the clustering technique k-means applied to the data in the two-dimensional space given by t-SNE. Using the gap [32] statistic and a Silhouette plot [33], the optimal number of clusters for the data set was estimated as 3. Despite testing other values of k, including k = 2, 4, 5, and 6, k = 3 provided a meaningful interpretation of the presence of STMPs in the signals. Before clustering, all the variables were transformed to a z score. 

In addition, differences between pairwise clusters were evaluated using the Fasano–Franceschini test [34] (*p* < 0.05) because the bivariate normal distribution assumption of the variables was violated (Kolmogorov–Smirnov test at the significance level of 5%).

#### 2.5.1. STMP Assessment

The frequency of the STMPs in each group, the transition probability among the STMPs in each group, and the time duration of the STMPs in each group were all evaluated. 

The transition probability among the STMPs was calculated based on the transition probability matrix denoted as P, which contains all transition probabilities. Assuming the states are 1,2, …, r, then the state transition matrix is given by
$$P=\left[\begin{array}{ccc} p_{11} & \cdots & p_{1r}\\ \vdots & \ddots & \vdots \\ p_{r1} & \cdots & p_{rr} \end{array}\right]$$

The probability of being in the state i and reaching the state j is given by $p_{ij}$ (Equation (2)).
(2)$$p_{ij}=P_{r\;}\left\{X_{n}=j|\;X_{n-1}=i\right\}$$

Although Equation (2) provides the probability of being in the same state ($r_{ii}\;\mathrm{or}\;r_{jj}$), it does not provide the state’s duration (i.e., persistence). The persistence time or time duration is defined as the time of a sample to remain in the same STMP. Then, to estimate the time duration of the STMPs for each group, we calculated the persistence time defined as D (Equation (3)).
(3)$$D(i)=\frac{n_{i}}{r_{i}}\;\times \;0.06$$
where *i* is the STMP group, $n_{i}$ is the amount of the samples in *i*-th STMP group, and $r_{i}$ is the number of permanence blocks for the *i*-th STMP type. The permanence block is the set of samples consecutive of the same STMP (Figure 4). The constant 0.06 converts the value into milliseconds. In the case of $r_{i}=0$, we considered $D\left(i\right)=n_{i}$. 

Figure 4 illustrates how persistence time is calculated. We estimated the persistence time for the STMP group identified as 2. Two steps should be followed to calculate duration time in any STMP group. In this case for STMP 2:
The number of samples in the STMP 2 was $n_{2}=20$;

The number of permanence blocks for each transition for the STMP 2 was r=6.

Thus, we obtained the persistence time of 0.2 ms for STMP 2 (Figure 4).

#### 2.5.2. STMP Characterization

To characterize each STMP in terms of the extracted features, we calculated the average for all features for each STMP for the HC and PD groups. Since the distribution of the data did not fit a normal distribution (Shapiro–Wilk test, *p* < 0.05), the Wilcoxon test was performed to compare the difference between the groups (HC and PD), with a significance level of 95% (*p* < 0.05).

## 3. Results

### 3.1. Explorative Cluster Analysis

The results from the k-means clustering were obtained from the feature vectors estimated from the data in a lower dimensional space estimated by t-SNE. The gap statistic and a Silhouette plot [32] were employed to estimate the optimal number of clusters (k) (Figure 5). According to both methods, k = 3 allowed for the identification of three clusters, i.e., STMP templates. 

Furthermore, to confirm the findings above, we tested the differences between pairwise clusters through the Fasano–Franceschini test (*p* < 0.05) (Table 3).

The results presented in Table 3 showed that the differences among pairwise clusters were significant. The statistical results supported our decision that three clusters gave us an optimal solution.

### 3.2. Identification of STMPs

From the analysis of clusters above, we identified three distinct types of STMPs present in the tremor in PD individuals. The distribution of these STMPs according to their similarity and groups (HC or PD) is shown in Figure 6. 

Figure 6 highlights the difference among STMPs. For STMP 1, most STMPs are from individuals in the control group. For STMP 3, most STMPs are from individuals with PD. For STMP 2, both experimental groups presented STMPs with a similar frequency. 

### 3.3. STMPs Assessment

The STMPs were assessed based on their frequency as well as the transition probability underlying their occurrences. Figure 7 shows the frequency of STMPs according to their similarity (i.e., STMP 1, 2, or 3) and experimental groups (HC and PD).

As shown in Figure 7, STMP 1 is more similar to those of the control group. In comparison, approximately 80% of STMP 3 belonged to people with Parkinson’s disease. STMP 2 had an equivalence number of STMPs in both experimental groups. 

Figure 8 shows the typical STMPs for the HC and PD groups. Although the signals from individuals with PD contained more segments with STMP 3 (Figure 8D), those individuals whose tremor was moderate had STMP 1 and 2 (Figure 8C), and the time series was similar to that of healthy individuals (Figure 8B). 

Additionally, we evaluated the dynamics of the STMPs according to the transition probability of their occurrence and the time of permanence in the underlying states related to the distinct STMPs types.

For the assessment of the transition, we adopted the transition probability matrix to calculate the probability of one STMP to transit to another one (Figure 9).

As presented in Figure 9, the HC group presented the highest transition values, especially from the 2 to 1 and 3 to 2 STMP types, whereas the PD group showed the lowest transition values. 

The permanence time in each STMP type was calculated according to Equation (3) for each group. Figure 10 illustrates the mean permanence time in ms of each STMP type. 

As shown in Figure 10, the PD group presented the highest mean permanence time (up to 10 ms in STMP 3). In contrast, the HC group presented about 3 ms in STMP 1. In STMP 2, both groups presented similar values of the mean permanence time, around 2.45 ms.

### 3.4. STMPs Characterization

The STMPs were characterized in terms of the extracted features. Table 4 shows the average for all the extracted features for both groups for each STMP type. Only CV and COMP were not significantly different (*p* > 0.05) between the HC and PD groups.

## 4. Discussion

Tremor characterization is critical for developing appropriate individualized treatment and rehabilitation strategies. According to our findings, there are at least three STMPs that are significantly different in tremor signals, and these STMPs can be characterized and evaluated based on amplitude, time of permanence and transition, and complexity. We found that the tremulous motion of the hand provided novel insights into the underlying temporal aspects of tremors that may be used as diagnostic and prognostic biomarkers for tremors.

The experimental protocol and signal preprocessing were based on the study from Andrade et al. [20]. The protocol used a pair of triaxial inertial units positioned on the back of the hand (Figure 2). Due to simplicity and the reported results in a previous study [20], we adopted the same protocol. However, only data from the gyroscope placed on the hand were used for analysis. In addition, to improve the data quality from gyroscope X, linear and nonlinear trends from the signals were removed following the same preprocessing steps described in [20]. As we investigated the short pieces of the signal, it is crucial to guarantee that small motions unrelated to the tremor motion and noise from the device are removed. 

The sensor and axis were chosen using a strategy based on the clustering of STMPs (Figure 6). Because the tremor manifests in all directions, we tested other sensors (e.g., an accelerometer and gyroscope) to determine which was best for detecting STMPs. The sensor performance based on the clustering results from gyroscope axis X (Figure 6) was more relevant for our purposes. Furthermore, in the setting shown in Figure 2, the gyroscope captured hand tremors better, particularly around the X-axis, where a tremor corresponding to small amplitude movements of wrist adduction/abduction (radial–ulnar deviation) and forearm pronation/supination occurs, and such movements are more pronounced than movements around the Y axis, which correspond to movements of wrist flexion/extension.

The main focus of most tremor studies is detecting or comparing the tremor of different groups/conditions and extracting features of the whole signal [10,20,26,27,35,36]. However, to identify underlying patterns in tremor signals, it is necessary to track them since tremors change over time. Thus, all signals were segmented with a 50% overlapping window to explore their dynamics [37]. For each segment, linear and nonlinear features were extracted (Table 2), as linear and nonlinear features provide complementary information for tremor assessment [38]. Morrison et al. [27] emphasized the importance of assessing the pattern of the tremor signal using nonlinear features. According to the authors, it is not always simple to discriminate different forms of tremor using only standard linear measures such as mean amplitude or the dispersion around a mean as traditionally defined. Table 4 highlights the importance of these feature combinations. It is possible to discriminate the groups using some of the features. 

The instantaneous fluctuations present in the signal contain physiologically meaningful patterns across multiple temporal–spatial scales. This study identified three (Figure 6) types of STMPs that were significantly different in the tremor signals (Table 3). T-SNE produced the visualization of STMPs (Figure 6), discriminating the different types of STMPs. Moreover, regarding discrimination among the STMP types, t-SNE was a relevant tool applied before executing the clustering step. Similarly, Oliveira et al. [39] found the highest accuracy in applying t-SNE as an a priori step to the classification. To our knowledge, this is the first study reporting cluster analysis techniques to examine STMPs in tremor signals using inertial sensors. This approach supported the concept that tremors are heterogeneous since we could identify a sequence of different STMPs in the tremor signal from a PD individual. Figure 7 shows that STMP 1 was present mainly in tremor signals from healthy people, although it was also present in tremors from PD individuals. Furthermore, the number of STMP 2s increased substantially in the PD individuals (Figure 8C), and STMP 3 was prevalent in PD individuals with a higher UPDRs score (Figure 8D).

Similarly, Dietz et al. [15] described that tremors in PD might be characterized by motor unit discharges. According to the authors, in those periods that the grouped discharges were more regular, the tremor amplitude decreased, and the firing frequency increased. Sometimes in these discharge periods, the tremor may disappear. It means that in a single tremor signal, it is possible to find distinct types of STMPs, as shown in Figure 8B,C. In addition, corroborating this result, Agapaki et al. [16] demonstrated the similarities between the characteristics and the activity of MUs in interspike intervals within doublets/triplets of tremors in PD and physiological tremors. 

Some signals from healthy people presented STMP 3, although about 80% of the signals from people with PD presented STMP 3. Comparing the results with the UPDRS scores of the PD group, signals containing a large number of STMP 1 were from PD individuals with a more moderate tremor. In contrast, those people from the PD group whose signals presented a large number of STMP 3s had a more severe tremor (Figure 8D). Surprisingly, some individuals in the HC group presented an expressive number of STMP 3s in their signals, possibly indicating the presence of a smoother tremor. A possible explanation for this result is that most people from HC are older adults, suggesting that tremors are related to aging, in which there exist segments of STMP 3 [40]. 

STMP 2 represents tremulous patterns associated with an intermediary state present for both groups in the same proportion. Interestingly, the appearance frequency of STMP2 may be related to tremor severity, as seen in the values of permanence time and transition probability (Figure 9 and Figure 10). STMP 2 is present in the signal of HC individuals all the time (Figure 9); thus, it has a higher mean probability of transition and lower permanence time in a single STMP (Figure 10). In contrast, the signal of PD individuals tends to have a lower transition probability and higher permanence time (Figure 5 and Figure 6). We may consider this higher transition probability from HC as variability, which provides adaptive strategies to maintain the control of tremors. Harbourne and Stergiou [41] emphasized the relevance of variability for maintaining health. They highlight that a lack of variability traps a behavior in a specific state or pattern, as shown in Figure 10 with the PD group. Individuals with PD tended to stay in state 3 related to STMP 3 for almost 30 ms (Figure 10), while healthy individuals stayed in state 1, related to STMP 1, for about 3 ms. Figure 8D shows a severe tremor; therefore, the whole signal had STMP 3. Then, we suggest that permanence time and transition probability in STMPs might be a source of behavioral change.

According to Harbourne and Stergiou [41], nonlinear tools best capture the hidden information in the variability, quantifying the structure of the signal. Sample entropy is a nonlinear tool used for measuring the degree of predictability or the structure of the variability of a time series. The difference among the STMPs of both groups was significant (Table 4, *p* < 0.05). For HC, the sample entropy values were higher than those for the PD group, indicating that the tremor was more regular and more predictable in the PD group. Similar results were found in studies where the tremor variability was around 15–22% lower in PD patients than in healthy controls, and the tremor was more regular in PD [25,26,27,42]. 

Furthermore, Gil et al. investigated entropy to evaluate hand tremors. They observed a 59.8% decrease in entropy for the resting tremor in PD patients and an increased tremor amplitude for these patients [43]. Similarly, Rissanen et al. [17] concluded that the EMG signals of PD individuals are more regular and contain more recurrent patterns than the EMG signals of healthy individuals. These previous findings supported our findings that the signals of individuals with PD are more regular (with less variability) than healthy individuals.

### Limitations

The sample size (N = 26) was determined according to the literature. Many studies evaluated tremors with similar samples size and reached significant results [16,20,22,26,27]. A greater sample size would validate the finding that tremors are heterogeneous, and more samples of each tremor severity may better characterize the signs of tremors.

## 5. Conclusions

This study described a method to identify and characterize STMPs in tremor signals. We assessed these STMPs from data obtained from a gyroscope placed on the dorsal side of the hand. We identified three STMPs in tremor signals and observed their prevalence depending on the tremor type, i.e., pathological or physiological, and individual condition. Moreover, we characterized these STMPs in terms of amplitude, permanence time, and complexity. The results confirm that tremors in individuals with PD tend to get trapped in a specific state. Therefore, this signal is less complex than the tremor in HC. 

In this sense, the methods used in this study can identify relevant landmarks for the follow up with tremor symptoms, assisting professionals in evaluating the tremor severity and the efficacy of a treatment.

## Figures and Tables

**Figure 1 healthcare-10-02536-f001:**
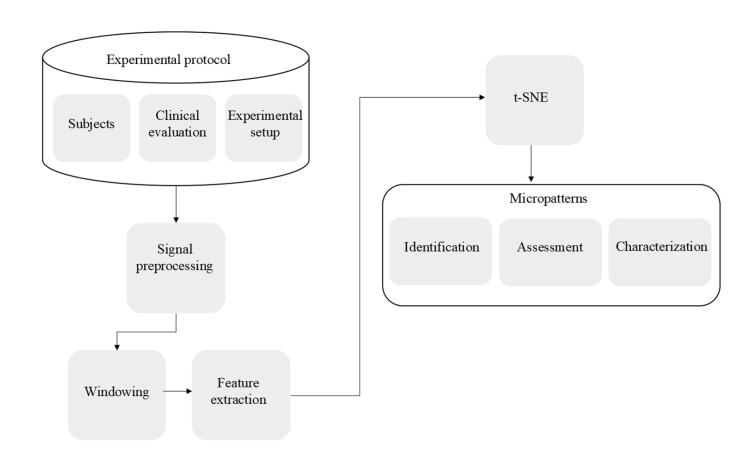
Diagram depicting the main steps for the identification and characterization of STMPs.

**Figure 2 healthcare-10-02536-f002:**
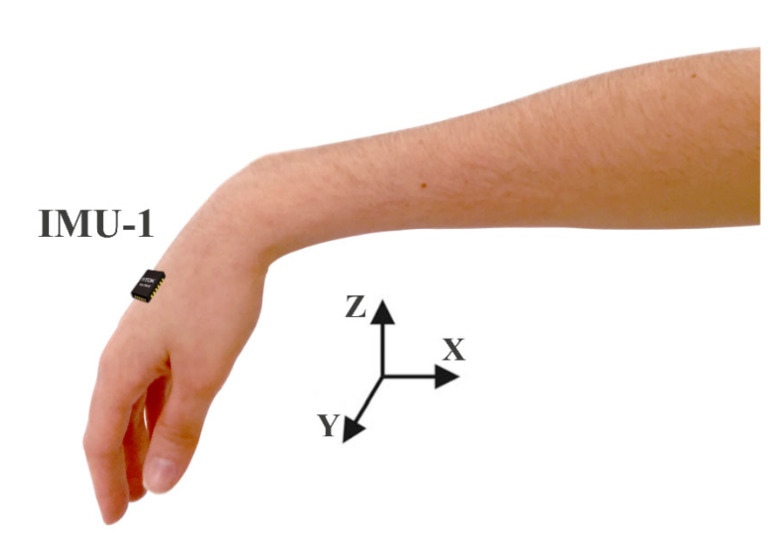
Illustration of the hand positioning during the rest tasks. The inertial sensor, IMU1, was placed on the dorsal side of the hand. X, Y, and Z are the proximal–distal, medial–lateral, and dorsal–palmar orientation axes, respectively.

**Figure 3 healthcare-10-02536-f003:**
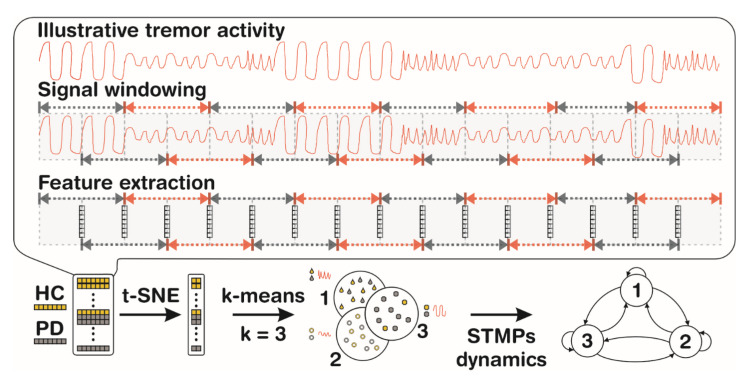
The main steps in tremor data analysis of healthy individuals (yellow) and individuals with PD (gray). The tremor activity may have distinct STMPs that emerge over time. The signal is windowed, and a feature vector is estimated for each overlapping window delimited by the arrows. Black and red colors are used to ease the visualization of the boundaries of each window. The set of features is estimated for individuals in the HC and PD groups. The high-dimensional data set is reduced to a lower-dimensional space using t-SNE, allowing the identification of clusters representing distinct STMP (represented by numbers 1, 2, and 3) templates present in the tremulous activity. Once these STMP groups have been identified, it is possible to understand their dynamics over time, i.e., the likelihood of STMP appearance, persistence, and regularity.

**Figure 4 healthcare-10-02536-f004:**
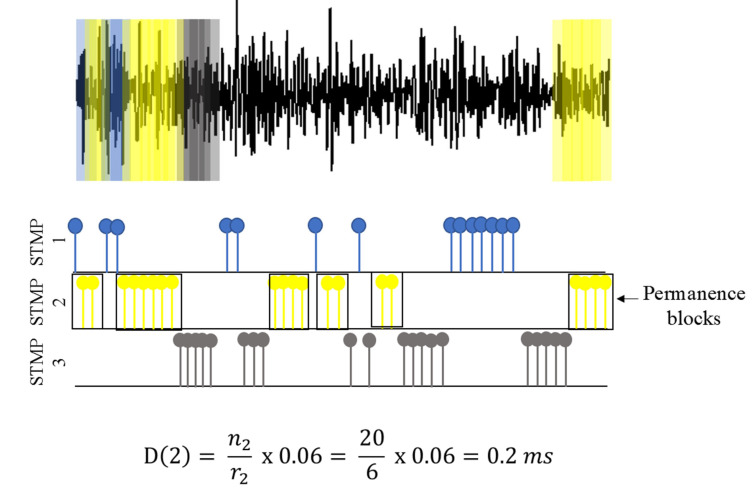
Calculation of the persistence time (Equation (3)) for the STMP 2. Three STMPs (represented by colors blue, yellow, and gray) were identified and distributed along the tremor signal and plotted below according to their appearance order. For STMP 2, the number of samples $n_{2}$ was 20, the number of permanence blocks was 6, and the calculated persistence time was 0.2 ms.

**Figure 5 healthcare-10-02536-f005:**
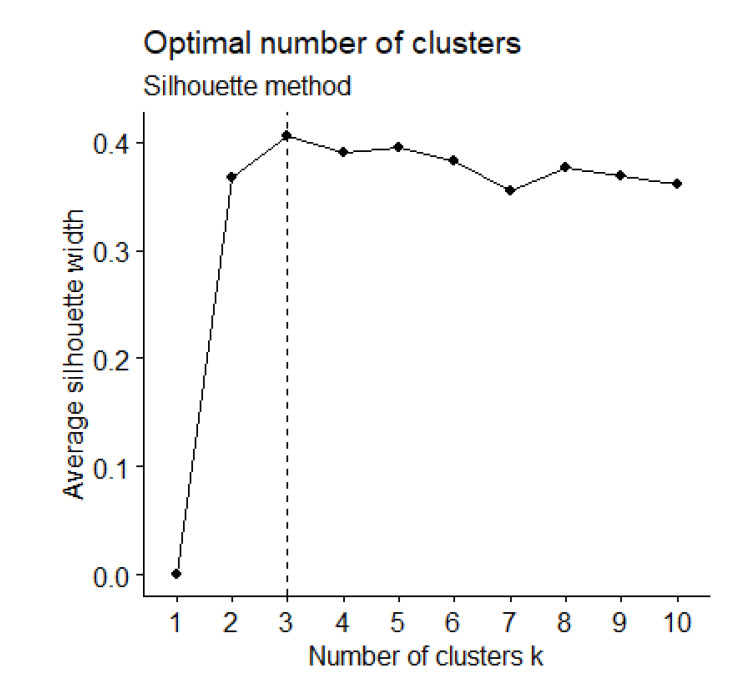
Silhouette plot indicating that the optimal number of clusters (k) equals to 3.

**Figure 6 healthcare-10-02536-f006:**
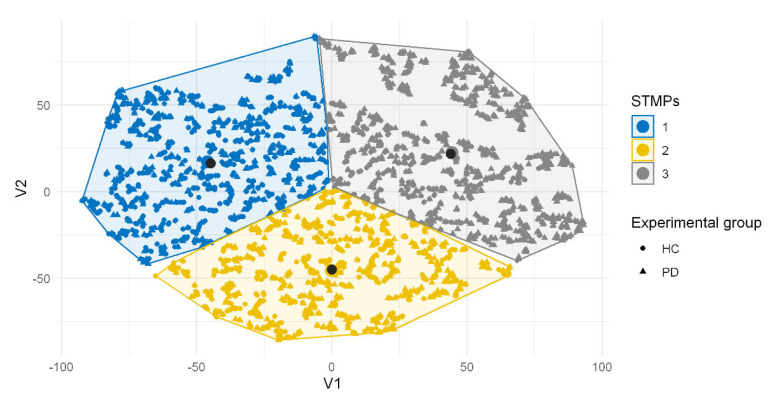
Distribution in three STMPs of all signal segments of both groups (HC and PD). The black circles highlight the cluster centers estimated by *k*−means. The STMP 1 (blue) has a predominance of individuals from control group. STMP 2 (yellow) has both experimental groups, while in STMP 3 (gray) most STMPs are from individuals with PD.

**Figure 7 healthcare-10-02536-f007:**
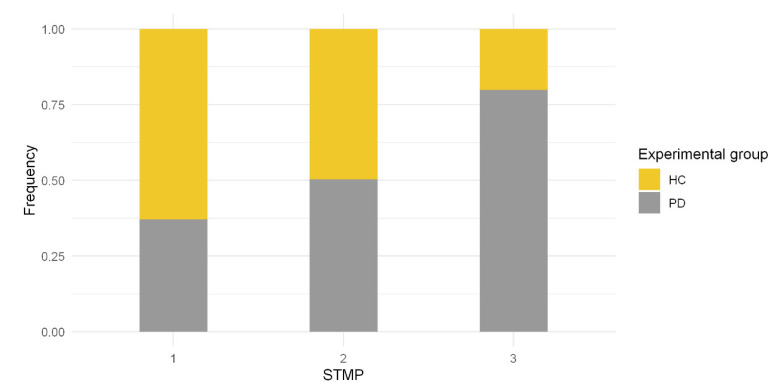
Frequency of STMPs based on each experimental group.

**Figure 8 healthcare-10-02536-f008:**
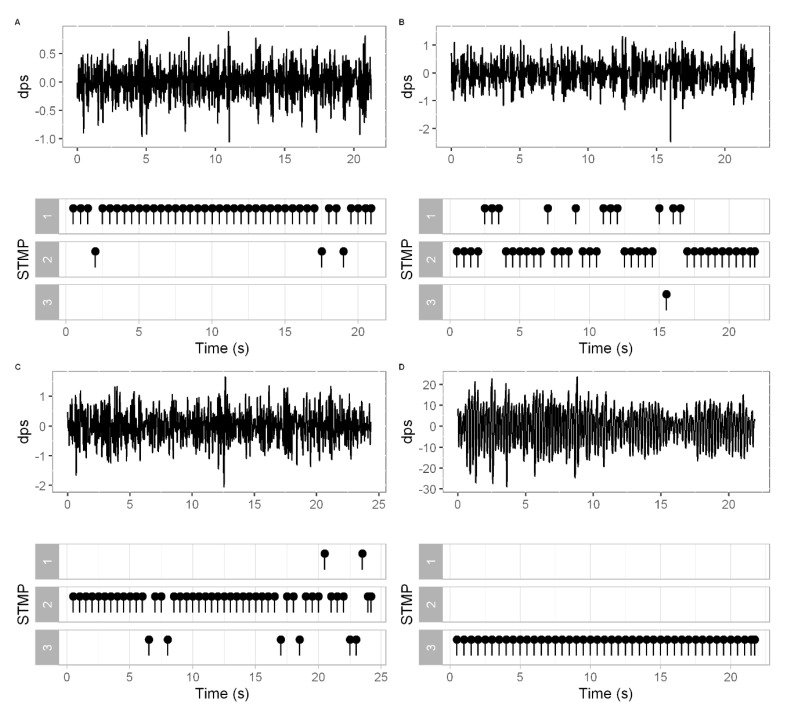
STMPs distributed along the tremor time series obtained from the gyroscope axis X. (**A**) Tremor signal from a healthy individual with the prevalence of STMP 1. (**B**) Tremor signal from a healthy individual with STMPs of all types. However, most of them are STMP 1 and 2. (**C**) Tremor signal from an individual with PD. Most of the STMPs are type 2. (**D**) Severe tremor signal from an individual with PD with the prevalence of STMPs type 3.

**Figure 9 healthcare-10-02536-f009:**
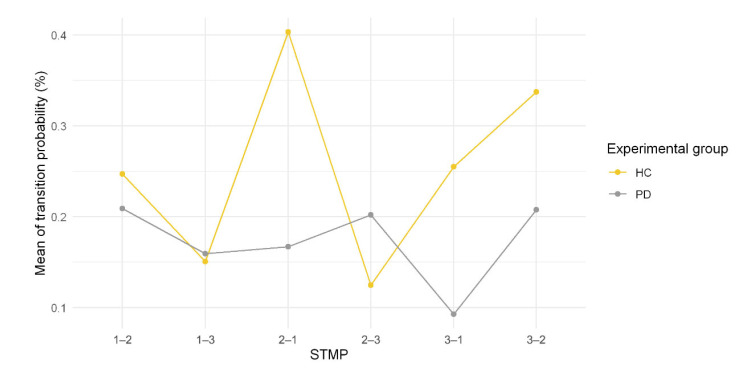
Transition probability between the STMP for each group (HC and PD).

**Figure 10 healthcare-10-02536-f010:**
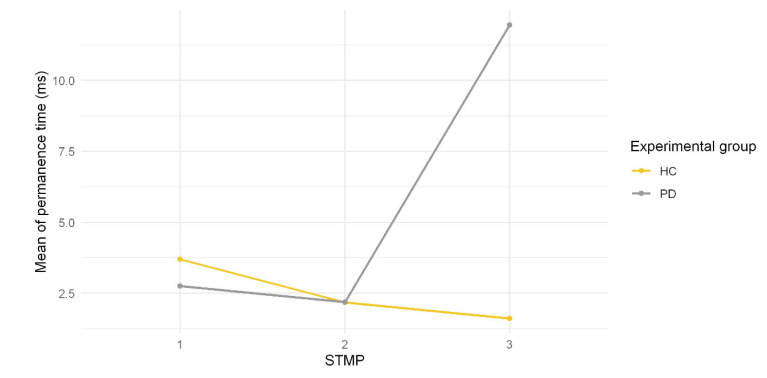
Mean of permanence time in each STMP for both groups.

**Table 1 healthcare-10-02536-t001:** UPDRS III scores for resting tremor on the most affected side of individuals with PD.

Subject Identifier	Age	Sex	Years Diagnosed with PD	Most Affected Side	Hand Tremor at Rest—Item 20 of the UPDRS
1	63	F	20	left	0
2	53	M	12	right	0
3	60	M	12	right	3
4	46	M	18	right	1
5	63	M	9	right	3
6	66	F	12	right	0
7	66	M	14	right	0
8	72	F	8	right	0
9	77	M	3	right	3
10	97	F	20	left	3
11	61	F	10	left	1
12	63	F	7	right	0
13	56	F	5	left	0
14	68	M	10	left	4
Mean ± SD	65.07 ± 11.98		11.42 ± 5.20		

**Table 2 healthcare-10-02536-t002:** Description of the features extracted from the signal. *N* is the total number of samples of the discrete time-series *x* of each window, *i* is the *i* discrete time instant, µ is the mean, and *σ* the standard deviation of *x*.

Feature	Description	Formula
MAV	Mean absolute value	$MAV=\frac{1}{N}\overset{N}{\underset{i=1}{\sum }}|x_{i}|$
CV	Coefficient of variation	$CV=\frac{\sigma \left(x\right)}{\text{µ}\left(x\right)}$
ZCR	Zero crossing rate	$ZCR=\frac{1}{2N}\overset{N}{\underset{i=1}{\sum }}|sgn\left(x_{i+1}\right)-sgn\left(x_{i}\right)|$ where $sgn\left(x_{i}\right)$ = $\left\{\begin{array}{c} 1,\;\;x_{i}\geq 0\\ -1,\;x_{i}<0\; \end{array}\right.$
SampEn	Sample entropy	$SampEn\;\left(m,\;r,N\right)=-\log\frac{A}{B}$ where *m* is the length of the template (length of the window of the different vector comparisons) and r is tolerance, which is usually selected as a factor of the standard deviation. For this application, we adopted *m* = 2 and *r* = 0.2 as in other studies [28,29]. *B* is the probability that two sequences are similar for *m* points, i.e., $d[X_{m}\left(i\right),\;X_{m}\left(j\right)]<r$, while *A* is the probability that two sequences are similar for *m* + 1 points, i.e., $\;d[X_{m+1}\left(i\right),\;X_{m+1}\left(j\right)]<r$ [30].
Hjorth parameters	Activity	$Activity={\;\sigma }^{2}\left(x\right)$
	Mobility	$Mobility=\sqrt{\frac{Activity\left(\dot{x}\right)}{Activity}}$ where $\dot{x}$ is the first discrete derivative of the x, i.e., $\dot{x}=\frac{x_{i}-x_{i-1}}{\Delta t}$ with temporal resolution $\Delta t=t_{i}-t_{i-1}$.
	Complexity	$Complexity=\sqrt{\frac{Mobility\left(\dot{x}\right)}{Mobility}}$
		where $\dot{x}$ is the first discrete derivative of the x, i.e., $\dot{x}=\frac{x_{i}-x_{i-1}}{\Delta t}$ with temporal resolution $\Delta t=t_{i}-t_{i-1}$.

**Table 3 healthcare-10-02536-t003:** Differences among pairwise clusters of STMPs given by the Fasano–Franceschini test.

Clusters	D-Statistic	*p*-Statistic
1 and 2	0.87	<0.05 *
1 and 3	0.97	<0.05 *
2 and 1	0.87	<0.05 *

Significant results for the differences between pairwise clusters are highlighted with “*”.

**Table 4 healthcare-10-02536-t004:** Mean of the extracted features for each STMP type for both groups.

Features	STMP	HC Group (Mean ± Std)	PD Group (Mean ± Std)	*p*-Value
MAV	1	0.27 ± 0.13	4.10 ± 31.81	<0.05 *
	2	0.32 ± 0.13	0.68 ± 1.12	<0.05 *
	3	0.28 ± 0.14	30.59 ± 45.21	<0.05 *
CV	1	82.71 ± 1128.10	861.72 ± 17,606.78	0.314
	2	−3.19 ± 435.55	−53.84 ± 517.96	0.078
	3	25.26 ± 240.14	14.37 ± 457.44	0.975
ZRC	1	6.32 × 10^−2^ ± 6.90 × 10^−3^	6.43 × 10^−2^ ± 1.01 × 10^−2^	<0.05 *
	2	5.15 × 10^−2^ ± 5.80 × 10^−3^	4.95 × 10^−2^ ± 5.90 × 10^−3^	<0.05 *
	3	4.80 × 10^−2^ ± 8.62 × 10^−3^	3.75 × 10^−2^ ± 1.06 × 10^−2^	<0.05 *
SampEn	1	0.52 ± 4.27×10^−2^	0.51 ± 6.40 × 10^−2^	<0.05 *
	2	0.48 ± 5.08×10^−2^	0.46 ± 5.12 × 10^−2^	<0.05 *
	3	0.46 ± 6.74×10^−2^	0.37 ± 8.80 × 10^−2^	<0.05 *
ACT	1	0.14 ± 0.19	1378.22 ± 11,695.60	<0.05 *
	2	0.19 ± 0.19	2.62 ± 18.21	<0.05 *
	3	0.16 ± 0.22	4000.02 ± 10,009.58	<0.05 *
MOB	1	0.20 ± 1.83 × 10^−2^	0.20 ± 2.85 × 10^−2^	<0.05 *
	2	0.17 ± 1.4 × 10^−2^	0.16 ± 1.55 × 10^−2^	<0.05 *
	3	0.16 ± 2.07 × 10^−2^	0.13 ± 2.85	<0.05 *
COMP	1	1.42 ± 0.14	1.45 ± 0.17	0.057
	2	1.39 ± 0.13	1.40 ± 0.13	0.284
	3	1.77 ± 0.18	1.67 ± 0.31	<0.05 *

* Wilcoxon test: significant difference between group.

## Data Availability

The data presented in this study are available upon request from the corresponding author. The data are not publicly available due to ethical restrictions.

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
