# Peer review of "Identification and Characterization of Short-Term Motor Patterns in Rest Tremor of Individuals with Parkinson’s Disease"

_healthcare, 2022, doi:10.3390/healthcare10122536_

Round 1

Reviewer 1 Report

In this work, the authors aim to identify and characterize short-term motor patterns (STMP) in parkinsonian resting tremor and physiological resting tremor. They recorded acceleration signals from participants with Parkinson’s disease (PD) and their healthy counterparts. Using a k-means clustering approach on reduced data features, they identified 3 STMPs in the tremor signal. They show that STMP 1 was prevalent in healthy control (HC) subjects, STMP 2 in both groups, and STMP3 in PD. Although an interesting take on characterizing STMP in the healthy and pathological states, the following concerns reduce my enthusiasm for the study:

1)      The authors do not clarify as to why they chose acceleration signals instead of EMG which would have more information on STMP.

2)      Furthermore, it is a well-known fact that resting tremor in PD is at a lower frequency compared to that of physiological tremor, but the authors never mention the frequency difference or attempt to check a time frequency decomposition of the tremor signals which in my opinion would be the most direct way of assessing tremor.

3)      Finally, I am not entirely sure about he motivation behind comparing STMP in the PD and HC subjects without including a formal classification performance using the identified STMPs. How is this superior to looking at either time domain or frequency domain features. In other works how does the clustering improve classification performance.

Another minor concern:

Most Figure legends would benefit by some additional description of the steps

Author Response

We thank you for the thoughtful and in-depth comments on our manuscript. Your suggestions and remarks have helped us reflect on the manuscript and make a better version. We carefully considered every comment you made and made the appropriate changes. All comments will be answered and complemented in the new manuscript, and some will be discussed here. Below we respond to your remarks on a point-by-point basis.

Yours Sincerely,

The authors

Reviewer 2 Report

The paper deals with a very interesting topic and attempts to find tremor patterns representative of PD patients. My main concern is that the k means clustering of Fig. 6 does not show 3 distinct clusters, i.e., I do not see clusters 1 and 2 being really different when compared to each other. Also, cluster 3 seems that it could be further divided into 2 sub-clusters as there is a gap zone traversing it. Therefore the question is if other cluster configurations would yield different results/conclusions.

Minor comments:

- In Table I: is it item 20 or 21?

- Line 149: why is there resampling?

- In Fig.3: the window overlap is not shown

Author Response

(The authors gave the same response as above.)

Round 2

Reviewer 1 Report

In response 1, more evidence is required for showing superiority of acceleration over surface EMG in detecting tremor bursts. A comparison showing how surface EMG is inferior in detecting tremor bursts is necessary. The claims made are not backed by data and/or citations. This would be believable if the authors show simultaneous EMG and gyroscope recordings from the same participant and make a comparison to prove why acceleration is better. 

In response 2, where are the PSD figures from?

Author Response

We thank you for the thoughtful and in-depth comments on our manuscript. Your suggestions and remarks have helped us reflect on the manuscript and make a better version. We carefully considered every comment you made and made the appropriate changes. All comments will be answered and complemented in the new manuscript, and some will be discussed here. 

Yours Sincerely,

The authors
